# Evaluation of Plasma Atherogenic Index, Triglyceride-Glucose Index and Other Lipid Ratios as Predictive Biomarkers of Coronary Artery Disease in Different Age Groups

**DOI:** 10.3390/diagnostics14141495

**Published:** 2024-07-11

**Authors:** Taha Okan, Mehmet Doruk, Ali Ozturk, Caner Topaloglu, Mustafa Dogdus, Mehmet Birhan Yilmaz

**Affiliations:** 1Kardiya Medical Center, 35000 Izmir, Turkey; 2Izmir Endocrinology Clinic, 35500 Izmir, Turkey; drmehmetdoruk@gmail.com; 3Department of Cardiology, Ozel Saglik International Hospital, 35000 Izmir, Turkey; aegeanaliozturk@gmail.com; 4Department of Cardiology, Faculty of Medicine, Izmir Economy University, 35550 Izmir, Turkey; topalolu@gmail.com (C.T.); mdogdus@hotmail.com (M.D.); 5Department of Cardiology, Faculty of Medicine, Dokuz Eylul University, 35220 Izmir, Turkey; prof.dr.mbyilmaz@gmail.com

**Keywords:** premature coronary artery disease, plasma atherogenic index, triglyceride-glucose index, lipoproteins ratios, coronary computed tomography angiography

## Abstract

(1) Background: Dyslipidaemia and insulin resistance are major risk factors for coronary artery disease (CAD). This study investigated the relationship between plasma atherogenic index (PA-I), triglyceride-glucose index (TGI) and other lipid ratios with the presence and prediction of CAD among different age categories. (2) Methods: The study included 223 participants diagnosed with CAD and those with normal coronary arteries (normal group) by coronary computed tomography angiography (CCTA). Participants were categorised by age and sex: premature CAD (PCAD) for men under 55 and women under 65, and older groups as elderly. (3) Results: PA-I, Lipid Combined Index, Castelli Risk Indices, and TGI were significantly higher in the PCAD group compared to the control group (*p* < 0.05). ROC analysis showed that a PA-I cut-off of 0.41 had a sensitivity of 62% and a specificity of 58% for predicting PCAD, while a TGI cut-off of 8.74 had a sensitivity of 68% and a specificity of 62%. In the elderly, no significant differences in these indices were found between the CAD and normal groups. (4) Conclusions: Traditional lipid profiles and non-traditional lipid indices such as PA-I and TGI show significant differences in predicting CAD in younger populations but not in older groups. TGI and PA-I may be promising biomarkers for the prediction of PAD, although further validation is needed.

## 1. Introduction

Coronary artery disease (CAD) is an important contributor to morbidities, accounting for 38% of cardiovascular disease (CVD)-related deaths in women and 44% in men [1]. The global prevalence of CAD has been increasing and is projected to reach 1845 cases per 100,000 population by 2030 [2]. It is imperative to use cardiac risk stratification to enhance therapeutic and preventative measures. Whether used alone or as part of prediction models, biomarkers are crucial to this process. A wide range of biomarkers associated with the risk of CAD have been discovered and examined in recent times. These include inflammatory markers like C-reactive protein (CRP), some of the interleukins, tumour necrosis factor-alpha (TNF-α), lipoprotein-associated phospholipase A2, lipoprotein-derived parameters, counts or ratios of monocytes and neutrophils, and others [3,4,5,6].

The master cause of CAD is atherosclerosis, which is impacted by hereditary and environmental variables such as oxidative stress, inflammation, and endothelial dysfunction. It is commonly known that there are risk factors linked to CAD. Dyslipidemia has been widely investigated and is significantly linked to the onset and progression of CAD. Specifically, it is marked by increased triglycerides (TG), low-density lipoprotein cholesterol (LDLC), decreased high-density lipoprotein cholesterol (HDLC), and total cholesterol (TC). Practically, LDLC reduction has traditionally been the main focus of lipid-lowering therapy [7,8,9]. However, even when LDLC is reduced to recommended levels, 50% of the remaining residual risk of CAD encouraged researchers to identify new predictors of CAD [10]. 

Compared to individual lipid parameters, comprehensive non-traditional lipid indices such as non-HDLC (total cholesterol minus HDLC), TC/HDLC (Castelli Risk Index-I), LDLC/HDLC (Castelli Risk Index-II), non-HDLC/HDLC (Atherogenic Index, AI), plasma atherogenic index (PA-I, the logarithm of the ratio of triglycerides to HDLC), and TC × TG × LDL/HDLC (Lipoprotein Combined Index, LCI) are considered superior predictors of CAD [11,12,13,14].

PA-I is based on the formula log(TG/HDLC) and is an indirect indicator of plasma atherogenicity. This is linked to the association between cholesterol esterification, residual lipoproteinemia and lipoprotein particle size [15,16]. It can accurately reflect the relationship between different lipoproteins and serves as an independent predictor of CAD in patients with suspected chronic coronary syndrome. PA-I may be a more discriminatory biomarker than traditional lipid parameters and other lipid ratios in predicting CAD [14,15,16,17]. As such, the Triglyceride/Glucose index (TGI) stands as a reliable indicator of insulin resistance (IR), which might exacerbate atherosclerosis through systemic inflammation, endothelial dysfunction and oxidative stress [17,18,19]. On the other hand, a higher TGI index was associated with an increased incidence of CAD or ACS compared to a lower TGI index. Furthermore, a possible linear dose-response relationship was found between the TGI index and the incidence of CAD/CVD. In a recent study, the TGI index and PA-I were shown to be linked to newly diagnosed CAD [16,17].

On the other hand, coronary computed tomography angiography (CCTA) is a non-invasive imaging modality that has a high sensitivity for the detection of CAD. The use of CCTA in the diagnostic evaluation of individuals with low to intermediate risk for obstructive CAD has been upgraded to a Class I recommendation in recent guidelines [20]. The primary method used in research investigating the relationship between different lipid indices and CAD has been the use of lumenographic criteria by invasive coronary angiography (ICA). However, recent studies suggest that CCTA is more accurate than ICA in assessing atherosclerotic burden [21,22]. Almost all studies that have examined the relationship between lipid-derived parameters and CAD have used ICA to diagnose CAD. In these studies, the CAD group typically consisted of individuals diagnosed with more than 50% coronary artery stenoses, while the control group consisted of individuals with less than 50% stenosis. However, it is important to note that patients with less than 50% stenosis or without luminal narrowing due to eccentric plaques with atherosclerosis may not accurately represent the normal group without coronary atherosclerosis, especially in studies conducted using ICA lumenographic criteria. The control groups in these earlier studies included patients with CAD, rendering the CAD prediction data problematic. This was related to the inclusion of individuals without CAD and those with less severe CAD jointly in the control groups in previous studies [11,12,13,14,23,24,25,26]. The aim of our study was to investigate the value of PA-I, TGI and other lipid-derived indices in determining the presence of CAD between individuals with truly normal coronary arteries, determined by CCTA and the patients with CCTA-confirmed CAD.

## 2. Materials and Methods

This is a pilot study. A cohort of 325 consecutive participants with chest pain were prospectively evaluated by CCTA between September 2023 and March 2024. Participants (1) have diabetes mellitus, (2) have atrial fibrillation, (3) with age < 35 years, (4) previous exposure to lipid-lowering drugs such as statins and fibrates, (5) having stage 3 or more chronic kidney disease, and (6) with known coronary or peripheral artery disease were excluded. Of note, individuals with a positive Treadmill/stress ECG test or myocardial perfusion scintigraphy for ischaemic heart disease (IHD) were sent directly to invasive diagnostic coronary angiography, as per guidelines recommendations and local practice, in an attempt to make differential diagnosis of obstructive epicardial disease from microvascular disease.

Hence, of the 325 participants, 223 were eligible for the study. In our study, premature coronary artery disease (PCAD) was defined as the onset of CAD before the age of 55 years in men and 65 years in women, as previously defined in the literature [8,27,28,29,30]. All participants provided baseline demographic information. Clinical and laboratory data were collected with patients’ consent and included sex, age, smoking history, height, weight, blood pressure, lipid values and family history of CAD. Blood samples were obtained from the antecubital vein in the early morning after an overnight fast of more than 8 h. Serum lipids and other biochemical parameters were measured using a biochemical analyser. HDLC and LDLC levels were measured by the homogeneous (direct) method, whereas TC and TG levels were measured by the enzymatic method. Written informed consent was obtained from all patients, and the study protocol was approved by the Ethics Committee of Bakırcay University. 

AI, PA-I, LCI, remnant lipoprotein cholesterol (RLP-C), Castelli risk index-I (CRI-I), Castelli risk index-II (CRI-II) were expressed in mmol/L and triglyceride/glucose index (TGI) was expressed in mg/dL and calculated by the formulae given in Table 1. Calculated according to the formulae;

CCTA was performed in all participants using a 128-slice single-source scanner (Somatom Go Top; Siemens Healthcare, Forchheim, Germany) and evaluated by an independent expert. Coronary artery calcification (CACS) was measured using the Agatston score and quantified with the same CT scanner.

Statistical Analysis: Continuous variables were expressed as mean ± standard deviation (SD) and compared using the independent samples *t*-test. Categorical variables were expressed as frequencies and percentages and compared using the appropriate chi-squared test. Correlations between PA-I, TGI and other variables were calculated using Pearson correlation analysis. Receiver operating characteristic (ROC) curves were constructed for PA-I and TGI to diagnose CAD in different age groups. All statistical analyses were performed using the Statistical Package for the Social Sciences (SPSS), version 29.0 (SPSS Inc., Chicago, IL, USA). A significance level of *p* < 0.05 in a two-tailed test was considered statistically significant.

## 3. Results

There were 123 participants who were candidates for PCAD and 100 participants with older age. Table 2 shows the baseline characteristics of participants with and without CCTA-confirmed CAD in the PCAD candidate group, comprising 123 participants, including men younger than 55 years and women younger than 65 years. In this group, male sex and smoking were significantly more common in the PCAD groups than in the true negative CCTA-excluded non-CAD control group. In addition, confirmed PCAD patients had significantly higher levels of TC, TG and LDLCs. In contrast, HDLC and non-HDLC levels were not significantly different in PCAD patients versus true negative non-CAD controls. In addition, the levels of non-traditional lipid indices, including TC/HDLC, LDLC/HDLC, AI, PA-I and triglyceride/glucose index, were significantly elevated in the PCAD group compared with the true negative non-CAD control group. In our study, PA-I and TGI showed the strongest statistical association with the presence of CAD in the PCAD candidate group. In our ROC analysis (Figure 1), the sensitivity of the cut-off value of 0.41 for PA-I in predicting PCAD was calculated to be 62%, with a specificity of 58%. Similarly, the sensitivity of the cut-off value of 8.74 for TGI in predicting PCAD was calculated to be 68%, with a specificity of 62% (Figure 2). Differences between the CAD and non-CAD groups in older age, defined as women aged 65 years and older and men aged 55 years and older, were shown in Table 3. In those older age groups, compared with the CCTA, which confirmed true negative non-CAD controls, the male sex was significantly more prevalent in the CAD groups, similar to the young group. However, in the older group, in contrast to the younger group, there were no statistically significant differences between CAD patients and non-CAD controls with CCTA-confirmed normal coronary arteries with regard to conventional lipid parameters such as TC, LDLC, HDLC, triglycerides and non-traditional lipid indices such as TC/HDLC, LDLC/HDLC, AI, PA-I and triglyceride/glucose index.

## 4. Discussion

This study focused on individual atherogenic particle-based differences between patients with CCTA-confirmed CAD and CAD-excluded true-negative controls according to the prematurity of CAD. Significant differences in some indices were found between patients with PCAD and true negative young control PCAD candidates with CCTA-diagnosed normal coronary arteries. However, these differences were not observed in the older age group.

Of note is that this study excluded participants with DM because there are significant differences in lipid metabolism and glucose levels between diabetics and non-diabetics. In particular, the HDLC, triglycerides and glucose levels affected by DM would directly confound the results of this study, as they are included in the numerator and denominator of the formulae of the TGI index, AI, PA-I and other indices [11,12,13,14]. Therefore, in our opinion, the exclusion of participants with diabetes was necessary.

Dyslipidaemia is a well-established risk factor for cardiovascular disease and plays a critical role in the development and progression of coronary atherosclerosis [8]. Because of the complex nature of lipoprotein metabolism, lipid ratios, and in particular indices like PA-I, which is derived from the logarithm of the molar ratio of TG to HDLC, are considered to be a more effective indicator of atherosclerosis than an assessment of individual lipid levels alone. PA-I is more sensitive to the interaction between atherogenic and protective lipoproteins [11,16,27].

Small dense LDLC (sdLDLC) is more easily infiltrated and deposited on the arterial wall due to its small particle size compared to LDLC and is readily oxidised to oxLDL. When oxLDL is phagocytised by macrophages, they transform into foam cells, exacerbating atherosclerosis and cardiovascular disease [31]. Several recent studies have identified sdLDL as an important marker for predicting atherosclerosis and recommended its clinical use. However, sdLDL detection is limited in clinical practice due to the complexity and high cost of the method. Previous research has shown an inverse relationship between PA-I and LDLC particle diameter, designating PA-I as a surrogate of sdLDL particle size. Thus, PA-I remains an inexpensive and feasible indicator of CAD [31,32,33,34]. On the other hand, triglyceride-rich lipoproteins (TRLs) are also involved in atherogenesis through several mechanisms. A previous study showed that elevated TRL was linked to low-grade inflammation, though an increase in LDLC was not. TRL and its remnants can increase the levels of cellular adhesion molecules in plasma, hence promoting monocyte adhesion to atherosclerotic lesions and producing an inflammatory cascade [35]. HDLC traditionally has important anti-inflammatory properties, such as regulating monocyte activation, preventing macrophage migration, and inhibiting LDLC oxidation. As a result, HDLC protects the arterial endothelium from the deleterious effects of LDLC [6,36]. Considering the mechanistic pathway of TG and HDLC in atherosclerosis, the PA-I, which is the logarithmic transformation of TG/HDL, might be a critical derived biomarker. In addition, the relationship between PA-I and sdLDL might provide another pathobiological link for CAD [32,33,34,35,36,37]. Patients with higher ratios of TC/HDLC, LDLC/HDLC, non-HDLC/HDLC, or combined ratios such as (TC × TG × LDLC)/HDLC are at increased cardiovascular risk due to an imbalance between highly atherogenic (artery clogging) lipoproteins and that carried by antiatherogenic (artery protecting) lipoproteins. This imbalance can result from either a higher proportion of atherogenic components in the numerator or a lower proportion of antiatherogenic components in the denominator [11,22]. 

Insulin resistance (IR) is central to the pathogenesis of both diabetes mellitus (DM) and coronary artery disease (CAD). It has been independently associated with vascular events in patients with and without DM, leading to coronary endothelial cell damage and necrosis [36,37,38]. IR also increases the formation of sdLDL, yielding Ox-LDL, which is a potent promoter of atherosclerosis. Besides, IR reduces the function of lipoprotein lipase, leading to increased triglyceride-rich lipoproteins and further atherosclerotic tendency [39,40]. The Homeostasis Model Assessment of Insulin Resistance (HOMA-IR) is currently the most widely used method for assessing IR. Recent research suggests that the novel, practical and convenient TGI index outperforms HOMA-IR in effectively assessing IR in both diabetic and non-diabetic individuals [18,41]. Recent research has shown that the TGI index is significantly linked to CAD [16]. It was an independent risk factor for the severity of PCAD and MACE [30]. In addition, the TGI index has been shown to be significantly associated with all-cause and cardiovascular mortality in the general population, particularly in those under 65 years of age [30,42,43,44,45], confirming our findings.

Premature CAD is a distinct subtype of CAD, typically characterised by onset before the age of 55 in men and 65 in women. Compared to CAD at a later age, people with PCAD have more severe disease with a higher incidence of acute coronary syndrome (ACS) due to unstable plaques and a higher incidence of major adverse cardiovascular events (MACE). This high rate of MACE persists despite the advent of drug-eluting stents, medically effective antiplatelet therapy, and aggressive lipid-lowering therapy [36,45]. Therefore, the prevention of coronary atherosclerosis, rather than the treatment of resulting MACE, is of paramount importance. 

Previous epidemiological studies designated that blood lipid levels alter with ageing, although the confounding role of age on blood lipid levels might significantly vary among different individuals. For example, one cohort found decreasing TG or LDLC levels in the elderly, whereas HDLC levels remained increased in this group. Another study in patients with AMI also showed that TG and TC levels were lower in elderly patients compared with young and middle-aged patients [46,47,48]. A recent study found an apparent decreasing trend in TG, TC, LDLC and PA-I levels with age in patients with CAD. However, this trend was less pronounced in patients without CAD. The researchers concluded that blood lipid levels change significantly with age in patients with CAD and that the association of PA-I with CAD was mainly found in the middle-aged population, aged between 35 and 64 years [23], which is in line with our findings.

In another recent study, the TGI index represented the highest relative risk among all of the biomarkers, analysed in participants with CVD less than 55 years, surpassing the associations of fasting blood glucose, lipids or inflammatory biomarkers. In the older patient group in the study, most clinical risk factors and cardiovascular risk biomarkers, such as traditional and non-traditional lipid profiles and CRP, showed attenuated age-related associations with CAD [48].

Our findings are consistent with these previous studies and provide additional evidence for the atherogenicity of serum lipids and non-traditional lipid-derived indices, particularly at young ages. Conventional lipid profiles and non-traditional indices such as AI, PA-I, CRI, CRII and TGI were significantly different in the PCAD group than in the control group with normal coronary arteries. In our study, the TGI index and PA-I show the strongest association with PCAD. In contrast, in the elderly cohort, there was no significant difference in conventional serum lipids levels and non-traditional lipid-derived indices between the CAD group and the control group with normal coronary arteries. 

Limitations: The size of our cohort is relatively small, and the findings represent a single-centre experience; hence, external validation is needed. Participants included in our cohort consisted only of individuals with relatively high lipid levels. Hence, the study is rather relatively biased and might not represent the overall population. According to our results, the sensitivity of the indices in the prediction of PCAD using the PAI and the TGI does not seem to be optimal. However, this is a pilot study and has the potential to open a new door for more comprehensive studies.

## 5. Conclusions

Conventional lipid profiles and non-traditional lipid indices derived from lipid parameters show statistically significant differences in men younger than 55 years and women younger than 65 years with premature CAD compared to those with CCTA-excluded true negative controls. However, in the older age group, there were no significant differences in these parameters between the CAD group and the normal coronary artery group. TGI and PA-I might emerge as derived biomarkers with acceptable sensitivity and specificity for predicting PCAD. They can be utilised in risk stratification for the early diagnosis of PCAD and may guide aggressive risk factor modification and therapy if needed. 

## Figures and Tables

**Figure 1 diagnostics-14-01495-f001:**
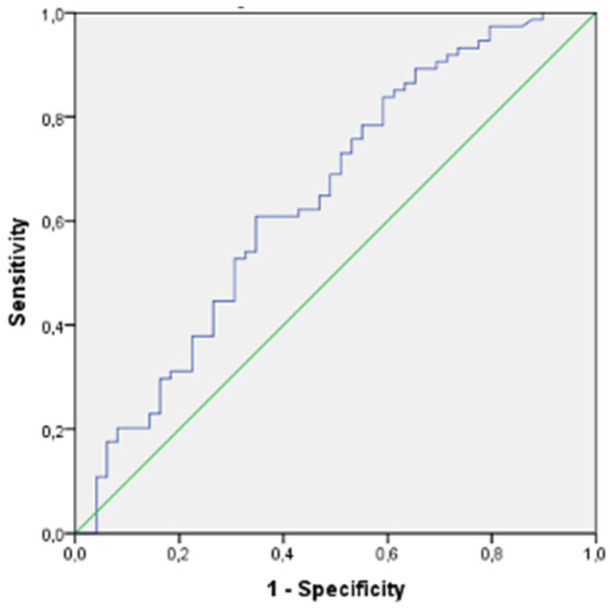
ROC curve of Plasma Atherogenic Index (PA-I) in the young group.

**Figure 2 diagnostics-14-01495-f002:**
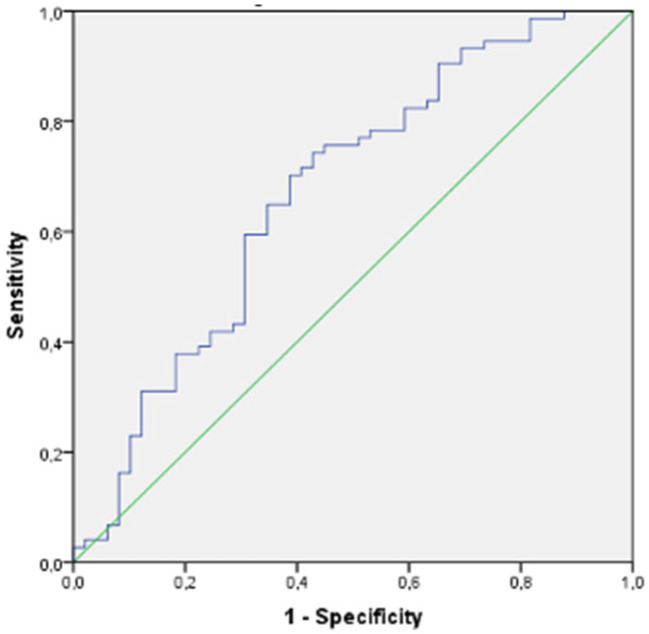
ROC curve of triglyceride/glucose index (TGI) in the young group.

**Table 1 diagnostics-14-01495-t001:** Formulae for indices derived from lipid parameters.

Index	Formulae
Atherogenic Index (AI)	Non-HDLC/HDL C
Plasma Atherogenic Index (PA-I)	log (TG/HDLC)
Triglyceride/Glucose index (TGI)	Ln (TG × glucose/2)
Castelli Risk index-I	TC/HDLC
Castelli Risk index-II	LDLC/HDLC
Triglyceride/Glucose index (TGI)	Ln (TG × glucose/2)
Lipoprotein Combined Index (LCI)	(TC × Tg × LDLC)/HDLC
Remnant Lipoprotein Cholesterol (RLPC)	TC − (HDLC) − (LDLC)
Non HDL Cholesterol	TC − HDLC

**Table 2 diagnostics-14-01495-t002:** Differences between the CAD group and non-CAD group for women < 65 years and men < 55 years.

Characteristics	Non-CAD Group *n* = 49	CAD Group *n* = 74	*p*-Value
Age (years)	54.12 ± 5.67	52.45 ± 5.86	0.121
Gender (female %)	29/49 (59%)	23/74 (31.1%)	0.003
Smoking	12/49 (24.5%)	33/74 (44.6%)	0.035
BMI (kg/m^2^)	26.7 8 ± 4.62	28.15 ± 3.92	0.080
AGATSTON	0	194 ± 310.17	0.001
Sytolic Blood Pressure (mmHg)	130.51 ± 15.09	133.89 ±14.93	0.65
Diastolic Blood Pressure (mmHg)	78.65 ± 9.50	82.45 ± 10.63	0.138
Fasting Blood Glucose (mmol/L)	5.48 ± 1.01	5.77 ± 0.82	0.407
Total Cholesterol (mmol/L)	12.16 ± 2.15	13.16 ± 3.19	0.040
Triglyceride (mmol/L)	8.23 ± 5.12	10.67 ± 5.63	0.016
LDLCholesterol (mmol/L)	7.45 ± 2.08	8.45 ± 2.84	0.037
HDL Cholesterol (mmol/L)	3.11 ± 1.01	2.97 ± 2.00	0.642
Non-HDL Cholesterol (mmol/L)	9.05 ± 2.27	10.19 ± 3.94	0.069
Total-C/HDLC (CR-I)	4.26 ± 1.43	4.99 ± 1.72	0.015
LDLC/HDLC (CR-II)	2.62 ± 1.14	3.17 ± 1.28	0.018
Non-HDLC/HDLC. (Atherogenic index)	3.26 ± 1.43	3.99 ± 1.72	0.015
Triglyceride/HDLC.	3.22 ± 2.97	4.19 ± 2.81	0.072
Lipoprotein combine index (LCI) = (TC × TG × LDL)/HDLC	334.92 ± 445.64	527.17 ± 498.61	0.031
Plasma Atherogenic index (PA-I) = log (TG/HDL C)	0.37 ± 0.33	0.53 ± 0.27	0.005
RLPC = TC − (HDLC) − (LDLC)	1.59 ± 1.43	1.74 ± 3.18	0.764
Triglyceride/Glucose Index = TGI = Ln (TG × glucose/2)	8.71 ± 0.64	9.05 ± 0.60	0.004

**Table 3 diagnostics-14-01495-t003:** Differences between the CAD group and non-CAD group for women ≥ 65 years and men ≥ 55 years.

Characteristics	Non-CAD Group *n* = 41	CAD Group *n* = 59	*p*-Value
Age (years)	65.63 ± 4.94	64.96 ± 5.56	0.121
Gender (female %)	22/41 (53.7%)	13/59 (22%)	0.001
Smoking	9/41 (22%)	20/59 (33.9%)	0.263
BMI (kg/m^2^)	29.12 ± 4.65	26.75 ± 3.26	0.004
AGATSTON	0	334.78 ± 405.05	0.001
Systolic Blood Pressure (mmHg)	136.78 ± 13.65	134.83 ± 17.09	0.545
Diastolic Blood Pressure (mmHg)	82.19 ± 11.18	79.01 ± 11.35	0.169
Fasting Blood Glucose (mmol/L)	5.55 ± 0.72	6.10 ± 0.85	0.166
Total Cholesterol (mmol/L)	12.68 ± 2.75	12.45 ± 2.48	0.665
Triglyceride (mmol/L)	9.02 ± 3.70	8.98 ± 4.98	0.970
LDLCholesterol (mmol/L)	8.26 ± 2.17	7.79 ± 2.22	0.298
HDL Cholesterol (mmol/L)	2.78 ± 0.58	2.82 ± 0.70	0.733
Non-HDLCholesterol (mmol/L)	9.89 ± 2.66	9.62 ± 2.38	0.589
Total-C/HDLC (CR-I)	4.71 ± 1.33	4.61 ± 1.32	0.708
LDLC/HDLC (CR-II)	3.08 ± 0.97	2.87 ± 0.93	0.282
Non-HDLC/HDLC. (Atherogenic index)	3.71 ± 1.33	3.61 ± 1.32	0.708
Triglyceride/HDLC.	3.55 ± 2.17	3.62 ± 2.87	0.903
Lipoprotein combined index (LCI) = (TC × TG × LDL)/HDLC	395.23 ± 318.88	369.22 ± 343.61	0.702
RLPC = TC − (HDLC) − (LDLC)	1.63 ± 1.32	1.82 ± 1.11	0.431
Plasma Atherogenic Index (PA-I) = log (TG/HDL C)	0.48 ± 0.22	0.45 ± 0.28	0.591
Triglyceride/GlucoseIndex = Ln (TG × glucose/2)	8.89 ± 0.44	8.92 ± 0.67	0.840

## Data Availability

Biochemical data and computerised tomographic coronary angiography images of all participants included in our study are available in the electronic archives of our clinics. In addition, SPSS files prepared for statistical purposes are also recorded. All data are available for access.

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
