# Peer review of "Evaluation of Plasma Atherogenic Index, Triglyceride-Glucose Index and Other Lipid Ratios as Predictive Biomarkers of Coronary Artery Disease in Different Age Groups"

_diagnostics, 2024, doi:10.3390/diagnostics14141495_

Round 1
Reviewer 1 Report
Comments and Suggestions for Authors
I am grateful to the editor for the opportunity to review the manuscript by Taha Okan et al, “Evaluation of Plasma Atherogenic Index, Triglyceride-Glucose Index and Other Lipid Ratios as Predictive Biomarkers of Coronary Artery Disease in Different Age Groups.”
In this article, the authors evaluate the capabilities of different biomarkers in identifying obstructive coronary artery disease. The difference of this publication is the verification of the presence of coronary artery disease using CT angiography of the coronary arteries, which made it possible to form a fairly adequate control group. As a result, the authors showed that the studied biomarkers were associated with CHD in young people, but not in the older age group.
During the review, I had the following questions and comments to which I would like to receive answers from the authors:
1. Verification of IHD is described sparingly in the article. Firstly, how was the presence or absence of microvascular ischemic heart disease verified? With this form, there may be no changes in CT angiography of the coronary arteries, but there will be positive functional tests. It is possible that the authors may have had such patients in the control group (according to experts, up to 20% of patients with coronary artery disease have microvascular angina, ref 1 – see below). In addition, calcification of the coronary arteries often makes it difficult to identify coronary stenoses. How did the authors of the article solve this problem?
2. I noticed very high lipid levels in this cohort of patients. Perhaps this depends on the local characteristics of the laboratory. Could the authors additionally indicate the reference values ​​of normal laboratory parameters for the studied biomarkers?
3. In section 2.1. Statistical analysis authors indicated that "Correlations between AIP and other variables were calculated using Pearson correlation analysis. Univariate and multivariate logistic regression analyzes were performed to examine the relationship between lipid parameters and the risk of CAD. The adjusted odds ratio (OR) per 1 SD increase in the corresponding lipid variable and 95% confidence intervals (95% CIs) were calculated." (lines 115-119). However, in the text of the manuscript, the authors did not provide data from either correlation analysis or logistic regression analysis.
4. The article does not have a section on Limitations of the study; it needs to be added.
References:
1. Merdler I, Chitturi KR, Chaturvedi A, Li J, Cellamare M, Ozturk ST, Sawant V, Ben-Dor I, Waksman R, Case BC, Hashim HD. Coronary microvascular dysfunction and inflammation: Insights from the Coronary Microvascular Disease Registry. Cardiovasc Revasc Med. 2024 May 11:S1553-8389(24)00488-3. doi: 10.1016/j.carrev.2024.05.020.
Comments on the Quality of English LanguageNo comments
Author Response
Comments 1:Verification of IHD is described sparingly in the article. Firstly, how was the presence or absence of microvascular ischemic heart disease verified? With this form, there may be no changes in CT angiography of the coronary arteries, but there will be positive functional tests. It is possible that the authors may have had such patients in the control group (according to experts, up to 20% of patients with coronary artery disease have microvascular angina, ref 1 – see below). In addition, calcification of the coronary arteries often makes it difficult to identify coronary stenoses. How did the authors of the article solve this problem?
Response 1:
Dear reviewer
As you said, CCTA does not have the competence to diagnose microvascular IHD. In our daily clinical practice, we recommend invasive coronary angiography for patients with a positive functional test for ischemia, as recommended by guidelines, since the likelihood of PCI or surgical treatment is high. Therefore, all individuals with normal coronary arteries who formed the control group of our study had negative functional tests or myocardial perfusion scintigraphy for ischemia. All our normal coronary artery group had an Agatston score of zero, i.e. they did not have coronary artery calcification. Since patients with coronary calcifications are already included in the CAD group, coronary calcifications do not pose a problem for the definition of the patient group and the normal group. With your very appropriate warning, the statement "Of note, individuals with a positive treadmill/stress ECG test or myocardial perfusion scintigraphy for ischemic heart disease (IHD) were sent directly to invasive diagnostic coronary angiography, according to guideline recommendations and local practice, in an attempt to make the differential diagnosis of obstructive epicardial disease from microvascular disease" was added to line 102-107 of the Material Methods section.
Comments 2: I noticed very high lipid levels in this cohort of patients. Perhaps this depends on the local characteristics of the laboratory. Could the authors additionally indicate the reference values ​​of normal laboratory parameters for the studied biomarkers?
Response 2:
Dear reviewer; All biochemical tests performed in our clinic are carried out in a laboratory with CAP, SEROCON, BIORAD, Oneworld Accuracy, LAB PT national and international external quality programme memberships. The reference values of the tests are universally compatible with the universally accepted values. The laboratory is ISO 9001-2000 certified and the ISO 15189 certification process of the laboratory is in progress. In conclusion, the results of the biochemical tests are reliable. Following your warning, we also found that the cholesterol levels of the people in our study were very high, as you said. Since the people enrolled in our study and assessed by CCTA had a high probability of having CAD, a cohort with high cholesterol levels may have been formed. With your very appropriate warning, we have added the following statement to lines 282-284 of the article in limitations section: "The participants included in our cohort consisted only of individuals with relatively high lipid levels, therefore the study is relatively biased and may not be representative of the overall population."
Comments 3:In section 2.1. Statistical analysis authors indicated that "Correlations between AIP and other variables were calculated using Pearson correlation analysis. Univariate and multivariate logistic regression analyzes were performed to examine the relationship between lipid parameters and the risk of CAD. The adjusted odds ratio (OR) per 1 SD increase in the corresponding lipid variable and 95% confidence intervals (95% CIs) were calculated." (lines 131-139). However, in the text of the manuscript, the authors did not provide data from either correlation analysis or logistic regression analysis.
Response 3:
Dear Reviewer; In accordance with your warning, we have corrected the methodological error we made in the statistical method without realising it; we have rewritten the statistics section as follows. Thank you for your warning. The rewritten statistical analysis section is below.
Statistical Analysis: Continuous variables were expressed as mean ± standard deviation (SD) and compared using the independent samples t-test. Categorical variables were expressed as frequencies and percentages and compared using the appropriate chi-squared test. Correlations between PA-I, TGI and other variables were calculated using Pearson correlation analysis. Receiver operating characteristic (ROC) curves were constructed for PA-I and TGI to diagnose CAD in different age groups. All statistical analyses were performed using the Statistical Package for the Social Sciences (SPSS), version 29.0 (SPSS Inc., Chicago, IL). A significance level of P < 0.05 in a two-tailed test was considered statistically significant.
Comments 4. The article does not have a section on Limitations of the study; it needs to be added.
Response 4: Dear Reviewer; A limitations section has been added to lines 275-281 of the article as follows, in accordance with your suggestion
Limitations: The size of our cohort is relatively small and the findings represent a single center experience, hence, external validation is needed. Participants included in our cohort consisted only of individuals with relatively high lipid levels, hence, the study is rather relatively biased and might not represent overall population. According to our results, the sensitivity of the indices in the prediction of PCAD using the PAI and the TGI does not seem to be optimal. However, this is a pilot study and has the potential to open a new door for more comprehensive studies.
Reviewer 2 Report
Comments and Suggestions for Authors Dear Editor This is an excellent research opportunity to find new diagnostic approaches. May I notify comments 1-The reported sensitivity of 62% for the cut-off value of 0.41 in predicting PCAD using the Personality Assessment Inventory (PAI) seems suboptimal. In biochemistry, higher sensitivity ( around 95%) is indeed desirable for robust diagnostic strategies. Given the lower sensitivity observed, additional recommendations or suggestions from the author would be valuable to enhance the diagnostic accuracy of PAI in this context . 2- I suppose author use more easily figure to presents the data 3-the manscrpit should accept after check for plagiarism checking and Ai, and proofreading 4- the author should give more attention towards introduction part 5- too many abbreviations, hard to read the text 6- the manuscript should introduce as A pilot study Comments on the Quality of English LanguageNone
Author Response
Comments 1:The reported sensitivity of 62% for the cut-off value of 0.41 in predicting PCAD using the Personality Assessment Inventory (PAI) seems suboptimal. In biochemistry, higher sensitivity ( around 95%) is indeed desirable for robust diagnostic strategies. Given the lower sensitivity observed, additional recommendations or suggestions from the author would be valuable to enhance the diagnostic accuracy of PAI in this context
Response 1: Dear reviewer;
The sensitivity and specificity of the P-AI and TGI indices we analysed are not very high, in line with your valuable opinion. As you mentioned, our study is a pilot study. It has the possibility to open a door for more comprehensive studies. We have also included your valuable opinion on this subject in the limitations section. line 275-281
Limitations: The size of our cohort is relatively small and the findings represent a single center experience, hence, external validation is needed. Participants included in our cohort consisted only of individuals with relatively high lipid levels, hence, the study is rather relatively biased and might not represent overall population. According to our results, the sensitivity of the indices in the prediction of PCAD using the PAI and the TGI does not seem to be optimal. However, this is a pilot study and has the potential to open a new door for more comprehensive studies.
Comments 2: I suppose author use more easily figure to presents the data
Response 2: Dear reviewer
We have tried to present the data with a figure in the format you suggest, but due to the large amount of data we could not create a more understandable figure. However, we have presented our data in tables. Please specify if there is a specific figure you would like.
Comments 4: the author should give more attention towards introduction part
Response 4: Dear Reviewer
In the format you suggested, the introduction section has been made more detailed by adding new sources. As follows; (Line 34-95
1. Introduction
Coronary artery disease (CAD) is an important contributor of morbi-mortality, accounting for 38% of cardiovascular disease (CVD)-related deaths in women and 44% in men [1]. The global prevalence of CAD has been increasing and is projected to reach 1,845 cases per 100,000 population by 2030 [2]. It is imperative to use cardiac risk stratification to enhance therapeutic and preventative measures. Whether used alone or as parts of prediction models, biomarkers are crucial to this process. A wide range of biomarkers associated with the risk of CAD have been discovered and examined in recent times. These include inflammatory markers like C-reactive protein (CRP), some of the interleukins, tumour necrosis factor-alpha (TNF-α), lipoprotein-associated phospholipase A2, lipoprotein-derived parameters, counts or ratios of monocytes and neutrophils, and others [3-6].
The master cause of CAD is atherosclerosis, which is impacted by hereditary and environmental variables such as oxidative stress, inflammation, and endothelial dysfunction. It is commonly known that there are risk factors linked to CAD. Dyslipidemia has been widely investigated and is significantly linked to the onset and progression of CAD. In specifically, it is marked by increased triglycerides (TG), low-density lipoprotein cholesterol (LDLC), decreased high density lipoprotein cholesterol (HDLC), and total cholesterol (TC). Practically, LDLC reduction has traditionally been the main focus of lipid lowering therapy [7-9]. However, even when LDLC is reduced to recommended levels, 50% of the remaining residual risk of CAD encouraged researchers to identify new predictors of CAD [10].
Compared to individual lipid parameters, comprehensive non-traditional lipid indices such as non-HDLC (total cholesterol minus HDLC), TC/HDLC (Castelli Risk Index-I), LDLC/HDLC (Castelli Risk Index-II), non-HDLC/HDLC (Atherogenic Index, AI), plasma atherogenic index (PA-I, the logarithm of the ratio of triglycerides to HDLC), and TCxTGxLDL/HDLC (Lipoprotein Combined Index, LCI) are considered superior predictors of CAD [11-14].
PA-I is based on the the formula of log(TG/HDLC) and as an indirect indicator of plasma atherogenicity. This is linked to the association between cholesterol esterification, residual lipoproteinemia and lipoprotein particle size [15,16]. It can accurately reflect the relationship between different lipoproteins and serves as an independent predictor of CAD in patients with suspected chronic coronary syndrome. PA-I may be more discriminatory biomarker than traditional lipid parameters and other lipid ratios in predicting CAD [14-16, 28]. As such, the Triglyceride/Glucose index (TGI ) stands as a reliable indicator of insulin resistance (IR) which might exacerbate atherosclerosis through systemic inflammation, endothelial dysfunction and oxidative stress [17-19]. On the other hand, a higher TGI index was associated with an increased incidence of CAD or ACS compared to a lower TGI index. Furthermore, a possible linear dose-response relationship was found between the TGI index and the incidence of CAD/CVD. In a recent study, the TGI index and PA-I were shown to be linked to newly diagnosed CAD [16,17].
On the other hand, coronary computed tomography angiography (CCTA) as a non-invasive imaging modality has a high sensitivity for the detection of CAD. The use of CCTA in the diagnostic evaluation of individuals with low to intermediate risk for obstructive CAD has been upgraded to a Class I recommendation in recent guidelines [20]. The primary method used in research investigating the relationship among different lipid indices and CAD has been the use of lumenographic criteria by invasive coronary angiography (ICA). However, recent studies suggest that CCTA is more accurate than ICA in assessing atherosclerotic burden[21,22]. Almost all studies that have examined the relationship between lipid-derived parameters and CAD have used ICA to diagnose CAD. In these studies, the CAD group typically consisted of individuals diagnosed with more than 50% coronary artery stenoses, while the control group consisted of individuals with less than 50% stenosis. However, it is important to note that patients with less than 50% stenosis or without luminal narrowing due to eccentric plaques with atherosclerosis may not accurately represent the normal group without coronary atherosclerosis, especially in studies conducted using ICA lumenographic criteria. The control groups in these earlier studies included patients with CAD, rendering the CAD prediction data problematic. This was related to the inclusion of individuals without CAD and those with less severe CAD jointly in the control groups in previous studies [11-14, 23-26]. The aim of our study was to investigate the value of PA-I, TGI and other lipid-derived indices in determining the presence of CAD between individuals with truly normal coronary arteries, determined by CCTA and the patients with CCTA confirmed CAD.
Comments 5: too many abbreviations, hard to read the text
Response 5: Dear Reviewer
You are right about the excessive use of abbreviations. As you say, the large number of abbreviations makes the article difficult to read and understand, but it is not possible to reduce the number of abbreviations due to the large number of lipid parameters examined. We have tried to overcome this difficulty by giving preference to commonly used and well-known abbreviations.
Comments 6: the manuscript should introduce as A pilot study
Response 6: Dear Reviewer
It is emphasised at the beginning of the material method section that the study as you suggest is a Pilot Study. ( line 97)
2. Materials and Methods
This is a pilot study. A cohort of 325 consecutive participants with chest pain were prospectively evaluated by CCTA between September 2023 and March 2024.
Reviewer 3 Report
Comments and Suggestions for Authors
Manuscript Title
Evaluation of Plasma Atherogenic Index, Triglyceride-Glucose Index and Other Lipid Ratios as Predictive Biomarkers of Coronary Artery Disease in Different Age Groups
Introduction
A separate para detailing the role and significance of AIP, TyG and other lipid-derived indices in CAD should be mentioned with recent citations.
Materials and Methods
Table should be introduced for detailing the clinical, biochemical and history of the included patients rather than a para.
Formulas should also be properly mentioned in a table.
Discussion
Section needs to be re-written. It should be crisp, well explanatory, and emphasize the importance of the discussed topic.
I would suggest a logical flowchart in the discussion elucidation the rationale of conventional lipid profiles and non-traditional lipid indices derived from lipid parameters in younger population with CAD compared to those with CCTA along with TyG and AIP as predictive biomarker for better explanation.
Thorough proof reading is recommended as in the Abstract authors have mentioned “ men 19 under 65 and women under 55…” and in the conclusion “ men younger than 55 years women younger than 65 years …..”. Kindly clarify
Conclusion: Revise the Conclusion section to offer a concise summary of key takeaways, emphasizing the importance of the discussed topics and potential clinical implications.
Limitation
This section should be introduced separately.
Minor editing of the English language is needed.
The manuscript will be attractive for the reader as it will improve our general understanding on CAD, which is a very important topic.
Comments on the Quality of English LanguageAuthor Response
Comments 1:I ntroduction
A separate para detailing the role and significance of AIP, TyG and other lipid-derived indices in CAD should be mentioned with recent citations.
Response 1:
Dear Reviewer; In line with your suggestion, we have added a separate paragraph to the introductory section explaining the role of P-AI and TGI indices in the aetiology and diagnosis of CAD. For the additional information, we have also referred to new studies published in 2023-2024, in line with your valuable suggestion. This paragraph, between lines 62-74 of the article, is given below.
Comments 1: Dear Reviewer; In line with your suggestion, we have added a separate paragraph to the introductory section explaining the role of P-AI and TGI indices in the aetiology and diagnosis of CAD. For the additional information, we have also referred to new studies published in 2023-2024, in line with your valuable suggestion. This paragraph, between lines 62-74 of the article, is given below.
PA-I is based on the the formula of log(TG/HDLC) and as an indirect indicator of plasma atherogenicity. This is linked to the association between cholesterol esterification, residual lipoproteinemia and lipoprotein particle size [15,16]. It can accurately reflect the relationship between different lipoproteins and serves as an independent predictor of CAD in patients with suspected chronic coronary syndrome. PA-I may be more discriminatory biomarker than traditional lipid parameters and other lipid ratios in predicting CAD [14-16, 28]. As such, the Triglyceride/Glucose index (TGI ) stands as a reliable indicator of insulin resistance (IR) which might exacerbate atherosclerosis through systemic inflammation, endothelial dysfunction and oxidative stress [17-19]. On the other hand, a higher TGI index was associated with an increased incidence of CAD or ACS compared to a lower TGI index. Furthermore, a possible linear dose-response relationship was found between the TGI index and the incidence of CAD/CVD. In a recent study, the TGI index and PA-I were shown to be linked to newly diagnosed CAD [16,17].
Comments 2: Materials and Methods: Table should be introduced for detailing the clinical, biochemical and history of the included patients rather than a para. Formulas should also be properly mentioned in a table.
Response 2
Dear reviewer
The calculation formulae you requested for the Materials and Methods section have been put into the tabular format you requested and tried to make them more understandable(Line 124-125)
The demographic and biochemical characteristics of the participants were already given in the tables organised for the results section.
Comments 3: DiscussionSection needs to be re-written. It should be crisp, well explanatory, and emphasize the importance of the discussed topic.I would suggest a logical flowchart in the discussion elucidation the rationale of conventional lipid profiles and non-traditional lipid indices derived from lipid parameters in younger population with CAD compared to those with CCTA along with TyG and AIP as predictive biomarker for better explanation.
Response 3:
Dear Reviewer, As you suggested, the discussion section has been rewritten in a more descriptive format. New references have been added. (Line 178-273) is given below
Discussion
This study focused on individual atherogenic particle-based differences between patients with CCTA-confirmed CAD and CAD-excluded true-negative controls according to the prematurity of CAD. Significant differences in some indices were found between patients with PCAD and true negative young control PCAD candidates with CCTA diagnosed normal coronary arteries. However, these differences were not observed in the older age group.
Of note, this study excluded participants with DM because there are significant differences in lipid metabolism and glucose levels between diabetics and non-diabetics. In particular, the HDLC, triglycerides and glucose levels, affected by DM would directly confound the results of this study, as they are included in the numerator and denominator of the formulae of the TGI index, AI, PA-I and other indices [11-14]. Therefore, in our opinion, the exclusion of participants with diabetes was necessary.
Dyslipidaemia is a well-established risk factor for cardiovascular disease and plays a critical role in the development and progression of coronary atherosclerosis [27]. Because of the complex nature of lipoprotein metabolism; lipid ratios, and in particular indices like PA-I, which is derived from the logarithm of the molar ratio of TG to HDLC, are considered to be a more effective indicator of atherosclerosis than assessment of individual lipid levels alone. PA-I is more sensitive to the interaction between atherogenic and protective lipoproteins [11,16,28].
Small dense LDLC (sdLDLC) is more easily infiltrated and deposited on the arterial wall due to its small particle size compared to LDLC, and is readily oxidised to oxLDL. When oxLDL is phagocytized by macrophages, they transform into foam cells, exacerbating atherosclerosis and cardiovascular disease [32]. Several recent studies have identified sdLDL as an important marker for predicting atherosclerosis and recommended its clinical use. However, sdLDL detection is limited in clinical practice due to the complexity and high cost of the method. Previous research has shown an inverse relationship between PA-I and LDLC particle diameter, designating PA-I as a surrogate of sdLDL particle size. Thus, PA-I remains an inexpensive and feasible indicator of CAD [32-35]. On the other hand, triglyceride-rich lipoproteins (TRLs) are also involved in atherogenesis through several mechanisms. A previous study showed that elevated TRL were linked to low-grade inflammation, though, LDLC increase was not. TRL and its remnants can increase the levels of cellular adhesion molecules in plasma, hence promoting monocyte adhesion to atherosclerotic lesions, and producing an inflammatory cascade [36]. HDLC traditionally has an important anti-inflammatory properties by regulating monocyte activation, preventing macrophage migration and inhibiting LDLC oxidation. As a result, HDLC protects the arterial endothelium from the deleterious effects of LDLC [6, 37]. Considering the mechanistic pathway of TG and HDLC in atherosclerosis, the PA-I, which is the logarithmic transformation of TG/HDL, might be a critical derived-biomarker. In addition, the relationship between PA-I and sdLDL might provide another pathobiological link for CAD [32-37]. Patients with higher ratios of TC/HDLC, LDLC/HDLC, non-HDLC/HDLC, or combined ratios such as (TC x TG x LDLC)/HDLC are at increased cardiovascular risk due to an imbalance between highly atherogenic (artery clogging) lipoproteins and that carried by antiatherogenic (artery protecting) lipoproteins. This imbalance can result from either a higher proportion of atherogenic components in the numerator or a lower proportion of antiatherogenic components in the denominator [11,22].
Insulin resistance (IR) is central to the pathogenesis of both diabetes mellitus (DM) and coronary artery disease (CAD). It has been independently associated with vascular events in patients with and without DM leading to coronary endothelial cell damage and necrosis [37-39]. IR also increases the formation of sdLDL, yielding Ox-LDL which is a potent promoter of atherosclerosis. Besides, IR reduces the function of lipoprotein lipase, leading to increased triglyceride-rich lipoproteins and further atherosclerotic tendency [40,41]. The Homeostasis Model Assessment of Insulin Resistance (HOMA-IR) is currently the most widely used method for assessing IR. Recent research suggests that the novel, practical and convenient TGI index outperforms HOMA-IR in effectively assessing IR in both diabetic and non-diabetic individuals [18,42]. Recent research has shown that the TGI index is significantly linked to CAD [16]. It was an independent risk factor for the severity of PAD and MACE [31]. In addition, the TGI index has been shown to be significantly associated with all-cause and cardiovascular mortality in the general population, particularly in those under 65 years of age [31,43-46] confirming our findings.
Premature CAD, is a distinct subtype of CAD, typically characterised by onset before the age of 55 in men and 65 in women. Compared to CAD at a later age, people with PCAD have more severe disease with a higher incidence of acute coronary syndrome (ACS) due to unstable plaques and a higher incidence of major adverse cardiovascular events (MACE). This high rate of MACE persists despite the advent of drug-eluting stents, medically effective antiplatelet therapy, and aggressive lipid-lowering therapy [37,46]. Therefore, the prevention of coronary atherosclerosis rather than the treatment of resulting MACE, is of paramount importance.
Previous epidemiological studies designated that blood lipid levels alter with aging, although the confounding role of age on blood lipid levels might significantly vary among different individuals. For example, one cohort found decreasing TG or LDLC levels in the elderly, whereas HDLC levels remained increased in this group. Another study in patients with AMI also showed that TG and TC levels were lower in elderly patients compared with young and middle-aged patients [47,50]. A recent study found an apparent decreasing trend in TG, TC, LDLC and PA-I levels with age in patients with CAD. However, this trend was less pronounced in patients without CAD. The researchers concluded that blood lipid levels change significantly with age in patients with CAD, and that the association of PA-I with CAD was mainly found in the middle-aged population, aged between 35 and 64 years [49] in line with our findings.
In another recent study, the TGI index represented the highest relative risk among all of the biomarkers, analysed in participants with CVD less than 55 years, surpassing the associations of fasting blood glucose, lipids or inflammatory biomarkers. In the older patient group in the study, most clinical risk factors and cardiovascular risk biomarkers, such as traditional and non-traditional lipid profiles and CRP, showed attenuated age-related associations with CAD [50].
Our findings are consistent with these previous studies and provide additional evidence for the atherogenicity of serum lipids and non-traditional lipid-derived indices particularly at young ages. Conventional lipid profiles and non-traditional indices such as AI, PA-I, CRI, CRII and TGI were significantly different in the PCAD group than in the control group with normal coronary arteries. In our study, TGI index and PA-I show the strongest association with PCAD. In contrast, in the elderly cohort, there was no significant difference in conventional serum lipids levels and non-traditional lipid-derived indices between the CAD group and the control group with normal coronary arteries.
Comments 4: Thorough proof reading is recommended as in the Abstract authors have mentioned “ men 19 under 65 and women under 55…” and in the conclusion “ men younger than 55 years women younger than 65 years …..”. Kindly clarify
Response 4: Dear Reviewer
The confusion in the two different sections you mentioned regarding the definition of men younger than 55 years and women younger than 65 years as PCAD has been eliminated and the correct definition has been included in both sections.
Comments 5: Conclusion: Revise the Conclusion section to offer a concise summary of key takeaways, emphasizing the importance of the discussed topics and potential clinical implications.
Response 5: Dear Reviewer, we have revised the Conclusion section as you suggested and tried to summarise our results and clinical implications. The revised Conclusion section is below. (Line 283-292)
Conclusions
Conventional lipid profiles and non-traditional lipid indices derived from lipid parameters show statistically significant differences in men younger than 55 years and women younger than 65 years with premature CAD compared to those with CCTA excluded true negative controls. However, in the older age group, there were no significant differences in these parameters between the CAD group and the normal coronary artery group. TGI and PA-I might emerge as derived-biomarkers with acceptable sensitivity and specificity for predicting PCAD. They can be utilized in risk stratification for the early diagnosis of PCAD and may guide aggressive risk factor modification and therapy if needed.
Comments 6: Limitation
This section should be introduced separately.
Response 6:
Dear Reviewer; A limitations section has been added to lines 275-281 of the article as follows, in accordance with your suggestion.
Limitations: The size of our cohort is relatively small and the findings represent a single center experience, hence, external validation is needed. Participants included in our cohort consisted only of individuals with relatively high lipid levels, hence, the study is rather relatively biased and might not represent overall population. According to our results, the sensitivity of the indices in the prediction of PCAD using the PAI and the TGI does not seem to be optimal. However, this is a pilot study and has the potential to open a new door for more comprehensive studies.
Comments 7:Minor editing of the English language is needed
Response 7: Dear Reviewer
We have revised the English of the entire article and tried to make it more fluent. We hope that the changes we have made will satisfy your request for a minor language revision.
Round 2
Reviewer 1 Report
Comments and Suggestions for Authors
The authors answered my questions and revised the text of the manuscript. I have no other comments.
Comments on the Quality of English LanguageNo comments